# Exploring family functioning and - hardiness in families' experiencing adult intensive care – A cross-sectional study

**Mona Ahlberg**[1,2]*, **Carina Persson**[3‡], **Carina Berterö**[2‡], **Susanna Ågren**[2,4]

**1** Department of Clinical Pharmacology, Linköping University, Linköping, Sweden, **2** Department of Health, Medicine and Caring Sciences, Linköping University, Linköping, Sweden, **3** Department of Health and Caring Sciences, Linnaeus University, Kalmar, Sweden, **4** Department of Cardiothoracic Surgery, Linköping University, Linköping, Sweden

☺ These authors contributed equally to this work.
‡ CP and CB also contributed equally to this work.
* mona.k.ahlberg@regionostergotland.se

**Data Availability Statement:** All relevant data are within the manuscript.

**Funding:** The present study is supported, in part, by a grant from the Health Research Council in the

## Abstract

Being cared for in an intensive care unit affects both the patient being cared for and the family in various ways. The family is of great importance for the recovery of the former intensive-care patient. The aim is to explore family functioning and family hardiness in families of former intensive care patients. A cross-sectional study using two self-reported questionnaires. Former adult intensive care patients and their family were recruited to participate between December 2017 and June 2019. The data were coded and entered the Statistical Package for the Social Sciences version 25, for analysis. To explore questionnaire data, descriptive and inferential statistical analyses were performed. Scale values were calculated on, both family wise and between the patients and the family members. STROBE checklist was used. Data was collected from 60 families (60 former intensive cared patients and 85 family members) and showed that 50 families scored healthy family functioning and 52 high strengths in hardiness. The data showed small variations between and within families for family functioning and family hardiness, there were only two families scoring low for both family functioning and hardiness. The variation was higher within the families, but there was no significance level.The conclusions were that family functioning and hardiness was, to a large extent, assessed as good by the families. Nevertheless, it is important to help the family obtain information and support. So, the family need to continue to communicate, finding coping abilities and strengths in adopting new strategies to protect the family unit. The family are very important for members' mental and physical recovery as the health of one family member affects the family as a unit.

## Introduction

Patients in adult intensive care units (ICUs) affected by critical illness, complications, or severe trauma, require close monitoring and advanced treatment [1]. Being critically ill and in need

South-East of Sweden (FORSS 466311). The funders had no role in study design, data collection and analysis, decision to publish, or preparation of the manuscript.

**Competing interests:** The authors have declared that no competing interests exist.

of intensive care can be stressful and life-changing for both the intensive care patient and their family [2, 3]. The experience of the ICU and critical illness affects the patient in particular, but also the family members and the family as a unit [4, 5]. The family is important for the patient's recovery and require information to be able to cope and give the patient the best support [6]. Identifying those families putting a greater access to those in need of more follow-up/information can increase mental and physical well-being of both the patient and their family members. Family functioning concerns the family's capabilities to develop and protect the family unit from major disruption, and enhance their coping ability and adaptation to changes [7, 8] Healthy family functioning is defined as a process of dynamically engaging with one another over time [7, 8] and unhealthy family functioning might occur within families, with high levels of conflict, disorganization, and poor affective and behavioral control [7, 8]. The individual development of family functioning only takes place in the context of significant emotional relationships [8]. Families need to relate to a "new" life together based on the often-harrowing event that an intensive care period entails, and in ICU and other contexts been shown to affect family functioning [9–11].

Family hardiness is the general atmosphere of interaction within the family and is seen as the family's resilience in other words coping abilities and strengths in adopting new strategies to protect the family unit, contributing to positive family functioning. Persons with low strengths in hardiness tend to feel alienated, powerless in the face of stressors, and tend to be more vegetative than vigorous in their approach to the changing events in their lives. This affects the family as a unit as all family members affect each other as literature overview of family hardiness in ICU and other contexts have shown [12, 13].

The family are the main providers of care in the home, supporting the ICU patient's rehabilitation and everyday life. As family affect each other within the family, it is important for health professionals to help the family to adapt or to support the family's adaptation, both during and after the ICU care [14–17].

Studies exploring important aspects of coping ability in families experiencing ICU could help find out which families are least able to cope with the situation of critical illness. This could help health professionals give more resources to these families. The aim was to explore family functioning and family hardiness in families of former intensive care patients.

## Materials and methods

### Design and methods

A quantitative cross-sectional design with validated self-rating scales was used to explore family functioning and family hardiness from patients' and family members' perspectives. The STROBE checklist has been used (See S1 Checklist).

**Setting and sample.** The former patients and their family were recruited at home after being cared for at four general ICUs in two regional—and two university hospitals in four cities in the south of Sweden, caring for patients having major or specialized surgery, respiratory failure, major trauma, sepsis, or requiring post cardiac arrest care.

**Data collection tools and methods.** The first author consecutively recruited the participants in collaboration with ICU administrators that listed from a database including former patients, participants eligible for inclusion. The inclusion criteria were: alive former ICU patients cared for ≥96 hours; age ≥18 years. The choice of ≥96 h was due to patient's severity of illness was shown to affect the family's ability to process the ICU care [18, 19]. The former ICU patient should be discharged one to two months before request, as literature shows that psychological symptoms, is highest during and near to ICU admittance [20]. Using

consecutive sampling, 390 participants fulfilling the inclusion criteria received a request to participate by means of an invitation letter, 60 answered and choose to participate.

In this study, 'family' was defined as the persons the family considered being family members, regardless of living in the same household or not [21, 22]. In the information letter, sent by mail, participants were told to decide which adult family members made up the family. After the consent form from the patient with information and addresses of the family members was returned, the information letter was sent to the family members recommended. There were no exclusion criteria, but the participants had to be able to understand and write Swedish. Recruitment was ongoing between December 2017 and June 2019.

**Data collection.** A package of questionnaires and a postage-paid return envelope were sent by mail after the consent forms were received and returned within the timeframe of one to three weeks. No reminders were sent.

**Questionnaires.** Data on the participants' characteristics were collected using a self-administered questionnaire. The questionnaires General Functioning Subscale; GFS [23, 24], was used to score family functioning and the Family Hardiness Index; FHI [25, 26] used to score the family hardiness. The participants were given instructions to complete the questionnaires individual.

The GFS based on the family systems theory to measure family functioning and validated to discriminate between healthy and unhealthy functioning. GFS was developed from the Family Assessment Device, FAD [24]. The GFS is a 12-item rating scale, measuring interaction within the family. The score can be calculated from one family member's responses or as a mean score of responses from several family members. Each item is rated on a four-point Likert scale: 'strongly agree = 1', 'agree = 2', 'disagree = 3' and 'strongly disagree = 4' [24]. The questionnaire was designed to assess family members´ individual perceptions of the family's ability to function, and includes six aspects: problem-solving, communication, roles, affective responsiveness, affective involvement, and behavior control. Examples of statements: 2. In times of crisis we can turn to each other for support. 4. Individuals are accepted for who they are. 6. We can express feelings to each other. 8. We feel accepted for what we are. The GF is a summative scale, where the total score is the mean of all items, divided in 12. Negative items are transformed before calculating the GFS score. Scores ≥2 denote perceived unhealthy family functioning. The Swedish version showed good reliability when tested in the context of participants who had undergone gastric bypass surgery (ordinal alpha = 0.9) [23]. In a Swedish study to investigate the outcomes of an intervention with the theory family systems nursing after ICU with the focus on family functioning and wellbeing the reliability coefficient Cronbach's alpha was 0.86 [18]. The reliability in our study had a Cronbach's α = 0.9.The FHI was used to measure family hardiness [25]. It is a 20-question Likert-type scale that measures family members´ individual perceptions of family stress resilience and ability to recover from life's hardships by changing their established patterns, to capture the healthy responses to stressful situations. Approach and attitude toward new experiences and the sense of being in control of family life protecting the family unit from major disruption, are measured. The questionnaire is scored on a four-point Likert-type scale: 'false' = 0, 'mostly false' = 1, 'mostly true' = 2, and 'true' = 3. Four subscales constitute the questionnaire: commitment, confidence, challenge, and control. Nine items are reversed before calculating the FHI score. Examples of statements are: 04. In the long run, the bad things that happen to us are balanced by good things that happen. 07. While we don't always agree, we can count on each other to stand by us in times of need. 11. We strive together and help each other no matter what. 15. We seem to encourage each other to try new things and experiences. 18. We work together to solve problems. The total score ranges between 0 and 60, with a higher score, national mean norm 47, reflecting higher family hardiness, better resilience [25]. The Swedish-translated instrument showed

good reliability, internal consistency for the FHI total scale was satisfactory, and the Cronbach's alpha = 0.8, tested in the context of family members of persons with cognitive dysfunctions [26]. In a study using FHI as one of the outcomes after trauma and surgery, in ICU showed α = 0.7 in reliability [19].

For the participants in our study, Cronbach's α = 0.8. There is no standard cut-off for this questionnaire, we used 1 SD as the cut-off as recommended by [27]. The mean score in this study was 47, with a standard deviation of 8, resulting in a cut-off score of 39.

**Data analysis.**   Power calculation based on data from a previous study using unpaired t-test. The calculation is based on a medium power size, 104 participants were needed.

$$ES \ = \ 0.6, \ \alpha \ = \ 0:05; \ 1 \ - \ \beta \ = \ 0.8$$

The data were coded and entered the Statistical Package for the Social Sciences (SPSS) version 25, for analysis. To describe the sample, univariate methods were used. Data were analyzed using both participants' characteristics and values from the questionnaires. The time of stay for the patient in the ICU was seen was an indicator of illness severity. Scale values from the FHI and GFS were accounted for, both family wise and between the former patients and family members in summary. To explore questionnaire data, descriptive and inferential (t-test and ANOVA) statistical analyses were performed. No corrections for missing data.

**Ethical and institutional approvals.**   The study was approved by the Ethics Review Board (record no. 2013/228-31, 2015/367-31, 2016/292-32, 2017/164-32, and 2018/572-32), and the research was carried out in line with the Declaration of Helsinki [28]. An information letter containing a presentation of the study, a description of the voluntary nature of the study, a consent form, and a postage-paid return envelope was sent to eligible participants and their family. All participant signed a consent form and sent it back. The first author dealt with all correspondence. The data was coded, and the code list was stored separately from the data [29].

The study was approved by the four Operations Manager in the different clinics and by the Ethics Review Board. The author got information about eligible participants matching the inclusion criteria from the administrators in the clinics. The administrator gave information about the days in intensive care and address to the patient. An information letter was sent to the patient describing: If you choose to participate, you are guaranteed confidentiality, which means that no outsiders will know who provided what information to the researcher. Everything that emerges from the study will be treated confidentially and no individual will be identifiable in the presentation of the study's results. All personal data is de-identified. You have the opportunity to cancel participation at any time during the study without giving a reason. Healthcare will not be affected whether you choose to participate in the study or not. The patient was then to inform which family members the author should/could contact for asking to participate and an information letter was sent to them. The written consent form was sent back by all participants in a pre-payed envelope, and then the questionnaires was sent with another pre-paid envelope.

## Results

The results showed that families, including the former ICU patient, from various families experiencing intensive care scored their family functioning and family hardiness as healthy. In total, data from 60 families (60 former ICU patients and their family members, 85; 145 participants) were analysed. Participants' and family members' characteristics are shown in Table 1 and family characteristics in Tables 1 and 2. In summary, the mean age of the former patients somewhat higher compared to the family members. Most former patients were males while the majority of family members were female. The participants were mostly pensioners or on

**Table 1. Sample characteristics of the former 60 ICU cared patients and 85 family members (n = 145).**

|  | Total family | ICU cared patients | family members | p-value |
|---|---|---|---|---|
| **Age, years; mean ±SD** | 59 ±16[a] | 64 ± 14 | 55 ± 17[a] | .001 |
| **Gender; female, n (%)** | 86 (59) | 25 (42) | 61 (72) | .000 |
| **Employment, n (%)** |  |  | [b] | .000 |
| Full time/part time |  | 17(28) | 52 (61) |  |
| Pension/disability pension/ sick leave |  | 43 (72) | 31 (36) |  |
| **Education, n (%)** |  | [c] | [d] | .000 |
| Elementary school or less |  | 19 (32) | 15 (18) |  |
| High school |  | 29 (48) | 33 (39) |  |
| University |  | 11 (18) | 34 (40) |  |
| **Days in the ICU for the patient; mean ± SD** |  | 13 ± 11 |  |  |
| **Measerments, mean (SD)** |  |  |  |  |
| FHI total scale | 47 (8)[b] | 48 (8) | 47 (8)[b] | .000 |
| GF | 1.5 (0.6) | 1.6 (0.6) | 1.4 (0.5) | .000 |

[a] Missing data 6 (7,1%)

[b] Missing data 2 (2,4%)

[c] Missing data 1 (1,7%)

[d] Missing data 3 (3,5%)

sick leave, but most family members worked full/part time. The majority, of the participants had a high school diploma (Table 1).

The former critically ill patient had an ICU stay of five to 70 days. The age of all the participants was 18 to 89. Most family members were partners/spouses (n = 47) and the family size was mostly two persons taking part in the study (n = 42) (Table 2).

The results of the t-test comparing the means (or averages) among ICU patients, family members and families to see if the groups differed, and the ANOVA exploring family functioning and family hardiness were non significant and are not reported.

According to GFS the cut-off for healthy family functioning is < 2 [24]. The mean value of GFS for all families was 1.5 ± SD 0.6. The results show that 50 families had healthy family functioning while ten families had low family functioning, with GFS ≥ 2. Among the families with unhealthy family functioning one family scored GFS 3, two families GFS 2.2, three families GFS 2.1 and four families GFS 2. The family members announced as friends, partners/spouses, children, and siblings with a mean age of 26 to 78 years. Days in ICU care ranged from 6 to 28 (Tables 1 and 2). The standard deviation of the data of family functioning compared between the participants and the family members was almost zero (Table 1). The SD within the families differed, 0 to 1.5 with 14 families having a SD above 0.6.

The FHI has a total score of 60, with a mean score for all families of 47, median 48 in this study. There were eight families with scores ≤ 39, which was used as cut-off in this study for low family hardiness. One family scored FHI 31, two scored FHI 36, one family had FHI 37, three families scored FHI at 38 and one family had FHI 39. The family members were announced as friends, spouses, and siblings with a mean age of 45 to 75 (Tables 1 and 2). The standard deviation of the data on family hardiness comparing the former patients and the family members was small (Table 1). The SD varied between 0 and 23 within the families, and as many as 19 families had an SD of 8 or above.

There were just two families having means scores of GFS ≥2 in combination with FHI scores ≤ 39, indicating that they experienced unhealthy family functioning and family hardiness. These were families with relation as friends and mean age was 45 and 57 respectively. The

**Table 2. Family characteristics and scale means of the GFS and the FHI of the 60 families.**

| Family members | Gender Woman | Age Mean | Relation | ICU care days | GFS mean ± SD | FHI mean ± SD |
|---|---|---|---|---|---|---|
| 2 | 1 | 73 | spouse | 5 | 1.1 ± 0.1 | 46 ± 5 |
| 3 | 3 | 38 | children | 15 | 1.2 ± 0.1 | 50 ± 4 |
| 2 | 2 | 56 | sibling | 14 | 1.2 ± 0.2 | 44 ± 1 |
| 4 | 3 | 56 | spouse/ children | 16 | 1.1 ± 0.1 | 50 ± 3 |
| 2 | 2 | 45 | friend | 6 | **3 ± 0.2** | **31 ± 23** |
| 2 | 1 | 60 | spouse | 6 | 1 ± 0 | 56 ± 0 |
| 3 | 3 | 41 | partner/child | 8 | **2.2 ± 0.7** | 42 ± 5 |
| 2 | 1 | 54 | spouse | 7 | 1 ± 0.1 | 53 ± 0 |
| 3 | 2 | 75 | spouse / child | 5 | 1.1 ± 0.1 | 47 ± 10 |
| 2 | 2 | 75 | spouse | 7 | 1.2 ± 0 | 57 ± 0 |
| 4 | 3 | 55 | spouse/ children | 7 | 1.5 ± 0.2 | 41 ± 12 |
| 2 | 1 | 74 | spouse | 9 | 1.2 ± 0 | 54 ± 1 |
| 2 | 2 | 74 | partner | 14 | 1.5 ± 0 | 42 ± 6 |
| 2 | 1 | 77 | spouse | 8 | 1 ± 0 | 48 ± 4 |
| 2 | 1 | 73 | spouse | 9 | 1.3 ± 0.1 | 47 ± 0 |
| 2 | 2 | 45 | spouse | 5 | 1 ± 0 | 52 ± 0 |
| 2 | 1 | 36 | partner | 7 | 1.3 ± 0 | 43 ± 1 |
| 2 | 1 | 73 | spouse | 14 | 1 ± 0 | **39 ± 12** |
| 3 | 2 | 51 | spouse/child | 8 | 1.7 ± 1.3 | 49 ± 12 |
| 2 | 1 | 66 | spouse | 7 | 1 ± 0 | 49 ± 8 |
| 2 | 1 | 69 | partner | 5 | 1.4 ± 0.6 | 48 ± 13 |
| 2 | 1 | 68 | partner | 70 | 1.4 ± 0.4 | 46 ± 3 |
| 4 | 3 | 62 | partner/children | 10 | 1.5 ± 0.5 | 50 ± 4 |
| 2 | 2 | 71 | partner | 10 | **2.1 ± 0.6** | 43 ± 14 |
| 3 | 2 | 73 | partner/child | 19 | **2 ± 1.1** | 46 ± 7 |
| 2 | 1 | 38 | spouse | 17 | 1.2 ± 0 | 55 ± 0 |
| 2 | 1 | 56 | sibling | 5 | 1.7 ± 0.4 | **38 ± 11** |
| 2 | 1 | 55 | partner | 8 | 1.2 ± 0.2 | 50 ± 1 |
| 2 | 1 | 72 | spouse | 10 | 1.2 ± 0.2 | 53 ± 1 |
| 4 | 3 | 54 | partner/children | 41 | 1.4 ± 0.4 | 52 ± 4 |
| 2 | 2 | 56 | spouse | 10 | 1.8 ± 1 | 41 ± 16 |
| 3 | 3 | 68 | friends | 10 | 1.2 ± 0.3 | 47 ± 3 |
| 5 | 1 | 51 | spouse/children | 23 | 1.5 ± 0.4 | 51 ± 3 |
| 3 | 1 | 51 | partner/child | 7 | 1.2 ± 0.5 | 42 ± 6 |
| 2 | 1 | 61 | spouse | 9 | 1.2 ± 0.2 | 41 ± 8 |
| 3 | 1 | 57 | friends | 15 | **2 ± 1** | **37 ± 1** |
| 2 | 1 | 46 | partner | 28 | **2.1 ± 0.1** | 50 ± 1 |
| 2 | 2 | 72 | spouse | 14 | 1.2 ± 0.1 | **36 ± 8** |
| 2 | 1 | 47 | sibling | 15 | 1.2 ± 0.2 | **36 ± 1** |
| 2 | 1 | 50 | spouse | 6 | **2.2 ± 0.1** | 44 ± 12 |
| 2 | 1 | 52 | partner | 7 | 1.2 ± 0.3 | 54 ± 1 |
| 2 | 1 | 67 | spouse | 6 | 1.1 ± 0.1 | 41 ± 1 |
| 2 | 1 | 73 | spouse | 8 | 1.3 ± 0.1 | 41 ± 1 |
| 2 | 1 | 48 | partner | 10 | 1.5 ± 0.4 | 54 ± 7 |
| 2 | 2 | 75 | friend | 6 | 1.3 ± 0.1 | **38 ± 5** |
| 4 | 1 | 37 | sibling | 30 | 1.9 ± 0.2 | 46 ± 14 |
| 2 | 1 | 66 | child | 13 | 1.1 ± 0.2 | 47 ± 1 |

*(Continued)*

**Table 2.** (Continued)

| Family members | Gender Woman | Age Mean | Relation | ICU care days | GFS mean ± SD | FHI mean ± SD |
|---|---|---|---|---|---|---|
| 2 | 1 | 63 | spouse | 6 | 1.5 ± 0.2 | 44 ± 13 |
| 3 | 1 | 74 | spouse/child | 9 | 1.9 ± 0.9 | 44 ± 13 |
| 3 | 1 | 48 | sibling | 7 | 1.9 ± 0.8 | **38 ± 0** |
| 2 | 1 | 78 | spouse | 10 | **2.1 ± 1.1** | 49 ± 1 |
| 2 | 1 | 26 | sibling | 19 | **2 ± 1.4** | 47 ± 8 |
| 3 | 1 | 51 | spouse/child | 37 | 1.8 ± 0.9 | 52 ± 3 |
| 2 | 1 | 77 | spouse | 7 | 1.8 ± 0.2 | 53 ± 8 |
| 2 | 1 | 64 | friend | 13 | **2 ± 1.5** | 55 ± 1 |
| 2 | 1 | 64 | spouse | 41 | 1.3 ± 0.5 | 43 ± 8 |
| 2 | 1 | 42 | spouse | 7 | 1.7 ± 0.2 | 53 ± 0 |
| 2 | 1 | 65 | spouse | 7 | 1.7 ± 0.1 | 52 ± 2 |
| 3 | 1 | 60 | spouse/child | 14 | 1.5 ± 0.2 | 50 ± 4 |
| 2 | 1 | 55 | spouse | 19 | 1.6 ± 0.6 | 47 ± 8 |

registered ICU stay was six and 15 days respectively. One of these families had a GFS score of 3 and FHI score of 31, and the other family had a GFS score of 2 and FHI score of 37 (Table 2).

## Discussion

The results show that 50 families had healthy family functioning while ten families scored family functioning, with GFS ≥ 2. Just 20% of the participants showed unhealthy family functioning, which corresponds to results from studies focusing on families experiencing various childhood diseases [9–11]. Compared to family functioning scores of a contemporary community sample mean score of 1.8 ± SD 0.4 [30], the results show that the families of former intensive care patients scored healthier functioning (1.5) but the variation was higher (± SD 0.6) within the group.

The family functioning in families not attending the study is unknown; probably families experiencing unhealthy family functioning did not participate to the same degree as families with better functioning, due to distress. A study found that family members experiencing the ICU are affected by it and have a significant burden [31].

The FHI has a total score of 60, with a mean score of 47 for the families in this study. The national mean score is 47 for the FHI [25] so the mean score of this study is fully comparable. There were eight families with scores ≤ 39, which was used as cut-off in this study. Somewhat lower mean scores were shown in a study comparing the strengths of hardiness between three groups of family members after a coronary artery bypass graft, after gunshot wounds or after a motor vehicle crash. The results show no significant differences, even if the family members of those suffering gunshot wounds reported the lowest mean score of ≈ 38, the mean of all the three groups were 42 ± 7 SD. The family in the study was bounded by biology, legal or social relationship [19]. Mutual influence among family members in other contexts has been shown to affect family hardiness [12, 13], and it is therefore reasonable to assume that this is also the case in the context of intensive care.

Two of 60 families scored unhealthy family functioning and family hardiness. The family's prior family functioning and family hardiness were not known in this study, important to acknowledge that all persons are individual and process their experience differently. As one study has shown, the need for follow-up has different timeframes and some need more follow-up than others [32] Studies have shown that families reporting increased symptoms of

depression and anxiety due to their experience of the ICU reported symptoms of these issues even prior to admission [33]. Several studies have reported positive intervention effects of family-centered care, and nurses' communication and psychosocial care were considered essential components of nursing interventions in the ICU [34, 35].

Which family members or participants may be of more risk of developing unhealthy family function and/or hardiness, could not be shown in this study. The two families that registered unhealthy family functioning and hardiness were friends and the only thing we know that differed about these families was that they did not live in the same household. A scoping review of families of patients in the ICU and their needs and satisfaction with care showed that information and the involvement of patients and family members are of importance [36]. Positive experiences of the ICU stay have been shown in a study which suggested caregivers' positive experiences may be associated with greater social support and better psychological well-being [37]. It is important to give the family and patient a feeling of belonging to the context. As healthcare professionals to give the family the capability to flourish and health to be able to maintain the patient's existing ability to rehabilitate after the critical illness and ICU [32].

## Limitations

A limitation of this study was that only 15% of the former ICU patients agreed to participate. Although power calculations show that medium size power was achieved, the low response rate might have led to an inaccurate positive picture. It seems resonable to presume that the most frail patients are not included in the study. Also of worth to consider is that low response rates produces bias only to the extent that there are differences between responders and non-responders on the estimate(s) of interest [37]. Due to ethical considerations non-responders estimates are unknown in the study. This response rate might have been higher if we had reminded the participants. Nevertheless, a consequence of the decision might be decreased variation within the sample since participants who hesitated to attend were probably among those having lower energy or less physical capacity to answer. We have not considered how ill the ICU patient was and in what condition he/she was at the time point for data collection, so this could be a limitation in the results, but we used the numbers of ICU care days, five to 70 days, as an indicator of illness in our analyses. The average length of stay is three days for ICU patients [1]. The lengths of stay are high, with mean 13 ± 11 days, which is above average, and this might strengthen the results. The fact that we have not reported if or how often the family members were present at the ICU to visit the patient and give support might be a limitation since support from health professionals an open conversation within the family is crucial for adaptation [38, 39]. The patient and family members in the same family scored differently, might be a sign of possible friction within the family and might benefit from family system nursing interventions [40].

The results may not be generalizable, and the accuracy should be confirmed by large-scale studies. This study was cross-sectional so the impact of family functioning and family hardiness outcomes over time remains unknown.

It is a strength that we had participants from four different hospitals and ICUs the variations of experience overall gave a good reflection of reality. Cronbach's alpha was used to support the validity of the sales in this specific sample. We demonstrated the value of the instruments in studies with similar families who have experienced similar situations in healthcare.

## Conclusion

The generalizable information, based on this study show healthy family functioning and hardiness in these families. The questionnaires we chose, determining groups of observations internally characterized by a high level of cohesion, both individual and familywise. Former ICU patients rated their family functioning and hardiness somewhat unhealthier compared to their family members. The standard deviation between the participants and the family members was high in some families and the fact that the patient and the family members scored differently might be a sign of friction within the family. This was a cross-sectional study exploring family functioning and hardiness and different designs and methods are needed to study the effect of intensive care on the family unit.

## Supporting information

**S1 Checklist. STROBE statement—checklist of items that should be included in reports of *cross-sectional studies.***
(RTF)

## Acknowledgments

The authors wish to thank the clinics for providing assistance in sample collection. The authors wish to extend their appreciation to the families experiencing critical illness and intensive care who generously shared their experiences with us. They also acknowledge the help and collaboration of the statistician and Senior Lecturer Mats Fredriksson at Forum Östergötland, Linköping University.

## Author Contributions

**Conceptualization:** Mona Ahlberg.

**Formal analysis:** Mona Ahlberg.

**Investigation:** Mona Ahlberg.

**Methodology:** Mona Ahlberg.

**Supervision:** Carina Persson, Carina Berterö, Susanna Ågren.

**Writing – original draft:** Mona Ahlberg.

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
