## [Decision Letter · Decision Letter 0]

28 Dec 2022

PONE-D-22-33585Exploring family functioning and - hardiness in families’ experiencing adult intensive care – A cross-sectional study.PLOS ONE

Dear Dr. Ahlberg,

Thank you for submitting your manuscript to PLOS ONE. After careful consideration, we feel that it has merit but does not fully meet PLOS ONE’s publication criteria as it currently stands. Therefore, we invite you to submit a revised version of the manuscript that addresses the points raised during the review process.

We look forward to receiving your revised manuscript.

Kind regards,

Wudneh Simegn Belay, MSc

Academic Editor

PLOS ONE

“The present study is supported, in part, by a grant from the Health Research Council in the South-East of Sweden (FORSS 466311).”

5. Please include your tables as part of your main manuscript and remove the individual files. Please note that supplementary tables (should remain/ be uploaded) as separate "supporting information" files

Reviewers' comments:

Reviewer's Responses to Questions

**Comments to the Author**

1. Is the manuscript technically sound, and do the data support the conclusions?

Reviewer #1: Partly

Reviewer #2: Yes

2. Has the statistical analysis been performed appropriately and rigorously? 

Reviewer #1: I Don't Know

Reviewer #2: Yes

3. Have the authors made all data underlying the findings in their manuscript fully available?

Reviewer #1: No

Reviewer #2: Yes

4. Is the manuscript presented in an intelligible fashion and written in standard English?

Reviewer #1: Yes

Reviewer #2: Yes

5. Review Comments to the Author

Reviewer #1: Dear authors,

thank you very much for the opportunity to review the submitted manuscript. The manuscript includes a very fine study about the hardiness of families of former ICU patients.

I am appreciating this work and have got some minor and serious concerns.

SERIOUS CONCERNS

Statistics: The GFS score is the mean of all 12 items, with a range between 1 and 4, correct? If the mean GFS was 1.5 (0.6), the result may not be normal distributed. Please check your data for distribution. In general, normal distributed data are reported as mean and standard deviation, and for metrical data tests such as t-tests etc can be used. Non-normal distributed data are reported as median and Interquartil Range (IQR), and other tests should be used for metrical data, such as Mann-Whitney U Test or other. Please, check the data for distribution and use appropriate tests. Please add this information to section data analysis, and – if needed- revise the results.

Same to FHI.

Tables: where are the tables???

MINOR CONCERNS

Section abstract is well written. Interestingly, it contains relevant information without reporting numbers (except for patients). Unusual, but fine.

Page 3, Line 49: The references are numbered, but “SIR, 2022” not. Please check.

Section background is fine. Main concepts are defined and referenced.

P4, L88 I wonder if the setting is correct? Maybe you recruited patients in this setting, but patients and families were surveyed at home, right?

P5, L93 to be clear “the participants” were former ICU patients? Patients were not recruited while being cared on an ICU, following the inclusion criteria, but recruited from a data base of former patients. If correct, you should revise add a term similar to “from a database including former patients”, or else.

Very good: these persons are not patients anymore, and you use the term “participants”. Excellent!

P5, L95: is that all inclusion criteria? Alive, ≥96h ICU, and ≥19y? What about cognitive capability to fill in a questionnaire, understand language, and others? Or having a family? What happened to those without family?

P5, L99: why 390 patients (patients or participants?)?

P5, L103: okay, to be clear: when persons (humans) are treated in hospital or go to the GP, they are becoming “patients”, including contracts, dependency, treatments etc etc etc. But after discharge, the contract is finished, and they are … persons, participants, citizens, men, women, humans again. With respect to the dignity of the persons and their families, and the non-ICU setting, these attributes should be clearly used. Please revise throughout the manuscript.

Section methods: I am impressed by the clear reports of the questionnaires. Well done.

P7, L160 “104”? What does it mean? Reference?

P10, L228 in the first paragraph in section discussion, the design and main results are reported, without adding new results, or references. Just a short overview for fast readers. Please revise.

Limits: you may shorten the section limitations to the core meaning.

Conclusions: conclusions are based on methods, results, discussion, and limitations. Conclusions are generalizable information, based on this study. Not on the “results” (line 303)

Line 303: check the grammar, the sentence seems not to be complete.

Reviewer #2: This study paper covers a significant and pertinent topic. An important and relevant topic is covered in this research work. Here are some comments to strength the research work.

Abstract:

Where is the study period?

Which type of software is used for analysis?

Regression results should be appeared in the result section

Introduction

What is the difference between introduction and background? Why you want to include both?

Method

Data collection tool and method

How many sample size you have?

How you select 60 families

You used a consecutive sampling and recruiting 390 patients that fulfill the inclusion criteria and they receive the request to participate by means of an invitation letter but in in your abstract section you write 60 patients and 85 families with a total of 145 participants.

Data analysis

How you calculate the power of your study? How it can be 104? How you calculate the power based on previous study? You should to calculate the power of your study?

Which type off software u used to calculate the power?

Results

In total you include 145 participants. Is this from 390 patients or not please clarify it and provide the response rate?

In your data analysis section you write as you perform an inferential statistics (t-test and ANOVA) but I haven’t seen any result of t-test and ANOVA?

6. PLOS authors have the option to publish the peer review history of their article (what does this mean?). If published, this will include your full peer review and any attached files.

Reviewer #1: No

Reviewer #2: **Yes: **Wondim Ayenew

---

## [Author Response · Author response to Decision Letter 0]

30 Jan 2023

Thank you for reviewing the previous manuscript, and for giving us the opportunity to send a revised manuscript giving a point to point response to the issues raised. We are grateful for the comments of the editors and reviewer. We have revised the manuscript in response to these helpful comments. A letter addressing the comments is submitted naming each concern listed by the reviewer and editors followed by description of specific changes, citing page and paragraph in the revision.

---

## [Decision Letter · Decision Letter 1]

12 May 2023

PONE-D-22-33585R1Exploring family functioning and - hardiness in families’ experiencing adult intensive care – A cross-sectional study.PLOS ONE

Dear Dr. Ahlberg,

Thank you for submitting your manuscript to PLOS ONE. After careful consideration, we feel that it has merit but does not fully meet PLOS ONE’s publication criteria as it currently stands. Therefore, we invite you to submit a revised version of the manuscript that addresses the points raised during the review process.

We look forward to receiving your revised manuscript.

Kind regards,

Wudneh Simegn Belay, MSc

Academic Editor

PLOS ONE

Reviewers' comments:

Reviewer's Responses to Questions

**Comments to the Author**

1. If the authors have adequately addressed your comments raised in a previous round of review and you feel that this manuscript is now acceptable for publication, you may indicate that here to bypass the “Comments to the Author” section, enter your conflict of interest statement in the “Confidential to Editor” section, and submit your "Accept" recommendation.

Reviewer #1: All comments have been addressed

Reviewer #2: All comments have been addressed

2. Is the manuscript technically sound, and do the data support the conclusions?

Reviewer #1: Yes

Reviewer #2: Yes

3. Has the statistical analysis been performed appropriately and rigorously? 

Reviewer #1: N/A

Reviewer #2: No

4. Have the authors made all data underlying the findings in their manuscript fully available?

Reviewer #1: Yes

Reviewer #2: Yes

5. Is the manuscript presented in an intelligible fashion and written in standard English?

Reviewer #1: Yes

Reviewer #2: Yes

6. Review Comments to the Author

Reviewer #1: Dear authors,

Thank you very much for the revised version of the manuscript. All concerns have been addressed, and you submitted a very fine manuscript. Well done, and thank you!

Reviewer #2: Thank you for considering my comments to strengthen your manuscript.

Comments

1. Better to include the software version you used in for analysis in the abstract section

2. Using consecutive sampling, you invited 390 participants. Then, 60 answered and chose to participate. In total, data from 60 families (60 former ICU patients and their family members, 85; 145 participants). The response rate will be 37.18%. How do you see it? Is it acceptable in research?

3. Table 1 and 2 describes the descriptive statistics only. They do not show the inferential statistics (t-test and ANOVA) but I haven’t seen any result of t-test and ANOVA? You should to do t-test and ANOVA analysis

7. PLOS authors have the option to publish the peer review history of their article (what does this mean?). If published, this will include your full peer review and any attached files.

Reviewer #1: **Yes: **Dr. Peter Nydahl

Reviewer #2: **Yes: **Wondim Ayenew Wudneh

---

## [Author Response · Author response to Decision Letter 1]

24 May 2023

Dear Reviewers Dr Peter Nydahl and Wondim Ayenew Wudneh,

Thank you for making the manuscript even clearer, I hope you find the adjustments correct. 

You can find the adjustments in the file reviewer comments.

With kind regards,

Mona Ahlberg

---

## [Decision Letter · Decision Letter 2]

20 Jun 2023

Exploring family functioning and - hardiness in families’ experiencing adult intensive care – A cross-sectional study.

PONE-D-22-33585R2

Dear Dr. Ahlberg,

We’re pleased to inform you that your manuscript has been judged scientifically suitable for publication and will be formally accepted for publication once it meets all outstanding technical requirements.

Kind regards,

Wudneh Simegn Belay, MSc

Academic Editor

PLOS ONE

Additional Editor Comments (optional):

Reviewers' comments:

Reviewer's Responses to Questions

**Comments to the Author**

1. If the authors have adequately addressed your comments raised in a previous round of review and you feel that this manuscript is now acceptable for publication, you may indicate that here to bypass the “Comments to the Author” section, enter your conflict of interest statement in the “Confidential to Editor” section, and submit your "Accept" recommendation.

Reviewer #2: All comments have been addressed

2. Is the manuscript technically sound, and do the data support the conclusions?

Reviewer #2: Yes

3. Has the statistical analysis been performed appropriately and rigorously? 

Reviewer #2: Yes

4. Have the authors made all data underlying the findings in their manuscript fully available?

Reviewer #2: Yes

5. Is the manuscript presented in an intelligible fashion and written in standard English?

Reviewer #2: Yes

6. Review Comments to the Author

Reviewer #2: Thank you for considering the comments to strengthen your manuscript.

All comments have been addressed.

7. PLOS authors have the option to publish the peer review history of their article (what does this mean?). If published, this will include your full peer review and any attached files.

Reviewer #2: **Yes: **Wondim Ayenew

---

## [Editor Report · Acceptance letter]

26 Jun 2023

PONE-D-22-33585R2 

Exploring family functioning and - hardiness in families’ experiencing adult intensive care – A cross-sectional study. 

Dear Dr. Ahlberg:

I'm pleased to inform you that your manuscript has been deemed suitable for publication in PLOS ONE. Congratulations! Your manuscript is now with our production department. 

Kind regards, 

on behalf of

Dr. Wudneh Simegn 

Academic Editor

PLOS ONE